# The Contribution of Multiplexing Single Cell RNA Sequencing in Acute Myeloid Leukemia

**DOI:** 10.3390/diseases11030096

**Published:** 2023-07-12

**Authors:** Lamia Madaci, Charlyne Gard, Sébastien Nin, Geoffroy Venton, Pascal Rihet, Denis Puthier, Béatrice Loriod, Régis Costello

**Affiliations:** 1TAGC, INSERM, UMR1090, Aix Marseille University, Parc Scientifique de Luminy, 13009 Marseille, France; lamia.madaci@inserm.fr (L.M.); sebastien.nin@inserm.fr (S.N.); geoffroy.venton@ap-hm.fr (G.V.); denis.puthier@univ-amu.fr (D.P.); beatrice.loriod@inserm.fr (B.L.); 2Hematology and Cellular Therapy Department, Conception Hospital, 13005 Marseille, France

**Keywords:** cell multiplexing, acute myeloid leukemia, single cell RNA sequencing, tumor heterogeneity, clonal evolution, targeted therapy, leukemia stem cell

## Abstract

Decades ago, the treatment for acute myeloid leukemia relied on cytarabine and anthracycline. However, advancements in medical research have introduced targeted therapies, initially employing monoclonal antibodies such as ant-CD52 and anti-CD123, and subsequently utilizing specific inhibitors that target molecular mutations like anti-IDH1, IDH2, or FLT3. The challenge lies in determining the role of these therapeutic options, considering the inherent tumor heterogeneity associated with leukemia diagnosis and the clonal drift that this type of tumor can undergo. Targeted drugs necessitate an examination of various therapeutic targets at the individual cell level rather than assessing the entire population. It is crucial to differentiate between the prognostic value and therapeutic potential of a specific molecular target, depending on whether it is found in a terminally differentiated cell with limited proliferative potential or a stem cell with robust capabilities for both proliferation and self-renewal. However, this cell-by-cell analysis is accompanied by several challenges. Firstly, the scientific aspect poses difficulties in comparing different single cell analysis experiments despite efforts to standardize the results through various techniques. Secondly, there are practical obstacles as each individual cell experiment incurs significant financial costs and consumes a substantial amount of time. A viable solution lies in the ability to process multiple samples simultaneously, which is a distinctive feature of the cell hashing technique. In this study, we demonstrate the applicability of the cell hashing technique for analyzing acute myeloid leukemia cells. By comparing it to standard single cell analysis, we establish a strong correlation in various parameters such as quality control, gene expression, and the analysis of leukemic blast markers in patients. Consequently, this technique holds the potential to become an integral part of the biological assessment of acute myeloid leukemia, contributing to the personalized and optimized management of the disease, particularly in the context of employing targeted therapies.

## 1. Background

Acute myeloid leukemia (AML) is a complex disease characterized by the infiltration of the bone marrow, blood, and other tissues by abnormal and often poorly differentiated cells of the hematopoietic system. This infiltration disrupts the normal production of blood cells, leading to the failure of normal hematopoiesis. AML exhibits genetic, epigenetic, and clinical heterogeneity, making it a challenging condition to treat effectively [1,2,3]. The disease predominantly affects the elderly population and is associated with severe complications and a high mortality rate, contributing significantly to cancer-related deaths. However, recent advancements in disease management have led to improved cure rates, reaching around 15% for patients over 60 years of age and approximately 40% for patients under 60 years of age [4]. Therefore, it is crucial to enhance the prognosis of AML by developing personalized treatment strategies that consider the initial heterogeneity of AMLs and account for potential changes over time, such as spatial and temporal heterogeneity of AML blasts and leukemia stem cells (LSCs) [5,6].

Advancements in molecular analysis techniques have significantly contributed to the treatment of hematological malignancies, particularly acute leukemias. Multiple factors play a role in tumor resistance to chemotherapy and immunotherapy [7]. Studying the overall expression of resistance markers, such as P-gp, MRP, GST, Bcl-2, TGFb, Gal-9, and CLIP, and identifying the metabolic pathways involved in resistance, has been of great interest [8,9,10,11,12,13,14]. However, analyzing the total population provides incomplete information, as it fails to identify the specific cell populations expressing these markers. The prognostic value of a resistance factor is likely to differ depending on whether it affects a terminally differentiated cell with low proliferation potential or a leukemic stem cell with high mitogenic potential, although this hypothesis is yet to be confirmed. Flow cytometry techniques have overcome some limitations by enabling the analysis of multiple markers on the same cell, but the number of markers is typically limited to around 50 and often less than 20 in routine tests. Mass spectrometry offers analysis of slightly over 100 parameters [15,16]. While this number of markers may seem substantial, it is still disproportionately small compared to the potential expression of approximately twenty thousand genes in a cell. This gap is bridged by current single cell RNA analysis techniques. Single cell RNA sequencing (scRNA-seq) has emerged as a powerful tool for investigating the intricate cellular transcriptome at a single cell level, especially in systems with cellular heterogeneity [17,18]. By combining high-throughput sequencing with bioinformatics tools, scRNA-seq enables the analysis of gene expression in multiple cell types within a tissue [19]. This technology facilitates the identification of rare and highly heterogeneous cell populations [20]. ScRNA-seq has found extensive application in various fields, including embryogenesis, developmental biology, immunity, neurology, and notably, oncology. It offers insights into both intra- and inter-tumor heterogeneity and the tumor microenvironment [21,22,23,24,25]. Several fascinating studies have been carried out on the use of the scRNA-seq technique in hematology, particularly in acute myeloid leukemia (AML) [26,27,28,29,30,31]. These studies have revealed in-depth characteristics of the bone marrow environment, providing a better understanding of AML [27]. In addition, they have highlighted the cellular hierarchy and transcriptional heterogeneity within AML, furthering our understanding of disease progression and the interactions between leukemic and immune cells [28].

Despite its effectiveness, classical single cell RNA sequencing has several drawbacks. The primary concern lies in cost, making it impractical for routine analysis. Additionally, comparing samples from different experiments proves challenging, even with standardized and normalized results. To address these limitations, there are several antibody-based techniques in scRNA-seq that allow for the simultaneous analysis of multiple samples, including Cell Hashing [32], CITE-seq [33], REAP-seq [34], and Abseq [35]. In our study, we have a particular interest in the Cell Hashing technique, which enables the identification and tracking of individual cells through the use of specific barcodes introduced prior to the scRNA-seq experiment. Initially introduced by Stoeckius et al. in 2018 [32], this method involves combining multiple samples and sequencing them together in a single tube. It relies on the labeling of cells with antibodies targeting a protein expressed in all cell types. These antibodies are coupled with hashtag oligonucleotides or HTOs, allowing for the differentiation of each sample after sequencing.

In 2022, Milazzo et al. demonstrated the applicability of cell hashing. They focused on investigating the effects of Panobinostat treatment on neuroblastoma cells and successfully identified the role of specific transcription factors in maintaining neuroblastoma cell proliferation [36]. In a separate study, Pankaew and colleagues employed cell hashing to examine the impact of MYC loss on the physiopathological development of PTEN-competent or PTEN-deficient T cells. Their findings indicate that the absence of MYC in CD4+ CD8+ thymocytes disrupts T cell homeostasis in the spleen, resulting in a significant reduction in the population of MYC-deficient effector/memory T cells [37].

The application of single cell hashing in acute myeloid leukemia (AML) presents an intriguing prospect, given the delicate nature of these cells. The material used, circulating leukemia cells, is readily obtainable, and accurately identifying the cell type is paramount—distinguishing differentiated blasts from leukemia stem cells and normal stem cells is crucial. In this study, our objective is to assess the viability and dependability of the single cell hashing technique on primary AML cells. To achieve this, we conducted a comparative analysis of the transcriptomic profiles of AML cells using the conventional single cell RNA sequencing method and the cell hashing technique.

## 2. Materials and Methods

### 2.1. Sample Preparation and Single Cell Preparation for Chromium Single Cell RNA Sequencing (scRNA-seq)

In this study, Human Peripheral Blood Mononuclear Cells (PBMCs) were obtained with the consent of AML patients (refer to Table 1). The isolation of PBMCs from peripheral blood involved centrifugation to separate them from plasma and other components of the blood, following a ficoll-Hystopaque density gradient centrifugation method adapted from English and Andersen [38]. Subsequently, the PBMCs were cryopreserved for long-term storage by freezing them either at −80 °C or in liquid nitrogen at −195.79 °C. To facilitate cryopreservation, a freezing medium comprising 90% FBS and 10% DMSO was used to store the PBMCs. Thawing of the cryopreserved cells in liquid nitrogen was performed at 37 °C, followed by rinsing with RPMI 1640 medium to eliminate the freezing medium containing DMSO. To enhance the sample quality by enriching it with viable cells, a Dead Cell Removal Kit (MACs Miltenyi Biotec, BG, Deustchland) was employed, following the manufacturer’s guidelines.

### 2.2. Single Cell RNA-seq Processing

For the implementation of the single cell RNA-seq technique, we followed the guidelines provided in the user guide for the “Chromium single cell 3’ reagent kits v3” from 10 × Genomics (cat # CG000183 Rev C). Additionally, for the Cell Hashing technique incorporating Feature Barcoding technology for Cell Surface Protein, we referred to the user guide for the “Chromium Next Gem Single Cell 3′ Reagent Kits v3.1” (cat # CG000206 Rev D).

#### 2.2.1. Cell Labeling with Cell Hashing Antibodies

Prior to cell labeling, a 50-μL aliquot containing 1,000,000 cells with over 95% viability was prepared for each sample. The cells were suspended in a Cell Staining Buffer (CSB) obtained from Biolegend (cat # 420201). The antibodies utilized for labeling were TotalSeq -A0251 (HTO1) and -A0252 (HTO2) antibody-oligo, sourced from Biolegend (cat # 394601/394603). To proceed with the labeling step, 1μg of HTO was added to each cell suspension. The two labeled samples were then incubated at 4 °C for 30 min. Subsequently, the cells underwent three consecutive washes with 3.5 mL of CSB, followed by centrifugation at 400 g for 5 min at 5 °C. Finally, the cells were resuspended in PBS (cat # 11503387) at a concentration of 1.106 cells/mL.

#### 2.2.2. Single Cell Library Generation

To prepare the single cell suspension, we employed the Single Cell 3′ Reagents Kit v3, following the guidelines provided in the Chromium Single Cell 3’ Reagent Kits User Guide (v3 Chemistry)—Library Prep—Single Cell Gene Expression—10x Genomics Support. Subsequently, the appropriate volume of each sample was diluted to recover 10,000 PBMCs. The single cell suspension, along with gel beads and oils, was then added to the 10 × Genomics single cell A chip. After generating droplets, the samples were transferred to PCR tubes for reverse transcription, which was performed using a C1000 Touch thermal cycler. Following reverse transcription, the cDNA was recovered and purified using silane DynaBeads, as specified in the user guide. Prior to cleanup with SPRIselect beads, we performed cDNA amplification for 12 cycles. The concentration of cDNA was measured using a Qubit 2.0 fluorometer (Invitrogen), while the quality of cDNAs was assessed through capillary electrophoresis using a fragment analyzer.

#### 2.2.3. Sequencing

For the preparation of libraries targeting HTO and gene expression sequencing, we utilized the NextSeq 500/550 High Output Kit v2.5 (75 cycles, Illumina) following the manufacturer’s recommendations. The library pool consisted of 8% HTO libraries and 92% gene expression libraries, and it was loaded onto the Flow Cell at a concentration of 1.8 pM. Subsequently, the libraries were sequenced on a Next Seq 500 system with the following parameters: 28 cycles for read 1, 8 cycles for index i7, and 91 cycles for read 2. This sequencing setup aimed to achieve an average sequencing depth of 32 million reads for HTO libraries and 368 million reads for gene expression libraries.

### 2.3. Bioinformatic Analysis

#### 2.3.1. Data Processing, Alignment, Gene Quantification and QA/QC

The sequencing data underwent analysis using the Cell Ranger Pipeline, specifically version 6.0.2, for various steps including quality control, sample demultiplexing using mkfastq, barcode processing, alignment, and single cell 3’ gene counting through cellranger count. Demultiplexing of samples was carried out with bcl2fastq, utilizing the 8-bp sample index, 10-bp UMI tags, and the 16-bp GemCode barcode. The alignment of the cDNA sequences presents in the 91-bp-long read 2 was performed using cell Ranger versions 3.0.0 and 3.1.0. UMI quantification, GEM-Code, and cell barcode filtering, based on Hamming distance error detection as described by Zheng et al., 2017 [39], were executed. Only confidently mapped transcriptome reads, excluding PCR duplicates, with valid barcodes and UMIs, were utilized to generate an unfiltered data matrix. Barcodes exceeding 10% of the 99th percentile of expected cell recovery (default = 3000) in total UMI counts were considered to contain cells and were selected to create a filtered gene-barcode matrix for subsequent analysis. The “gene and transcripts (UMI counts) per cell” metric was employed to compare the sensitivity of scRNA-seq. Additionally, the “Fraction Reads in Cells” parameter was calculated to assess the presence of background cell-free (ambient) RNA in the cell suspension, based on the fraction of cell-associated barcoded reads that confidently mapped to cell barcodes.

#### 2.3.2. Normalization and Correlation of Gene Expression Levels

Normalization was performed individually for each sample using the “log-normalization” method applied to the Seurat object. To assess the correlation between the various samples, an initial Seurat filter was employed to retain genes detected in at least three cells and to include cells with a minimum of 200 UMI. Subsequently, the gene expression correlations across different samples were calculated by determining the average expression of genes shared by all samples.

#### 2.3.3. PCA and tSNE Analysis for Cell Clustering and Classification, and Data Visualization

Secondary analysis was conducted using the Cell Ranger count and aggr pipelines. Prior to cell clustering, Principal Component Analysis (PCA) was applied to the gene-barcode matrix, which had been normalized, log-transformed, centered, and scaled, to reduce the dimensionality of the feature (gene) space. The implementation of the IRLBA algorithm in Python was employed for this purpose, resulting in a projection of each cell onto the first N principal components. No filtering was performed on “low-quality” genes and cells as previously described, and the Seurat package utilized this projection in the subsequent PCA analysis. Following PCA, t-distributed Stochastic Neighbor Embedding (t-SNE) was utilized to visualize the cells in a 2-D space. Clustering was then performed to group cells with similar expression profiles based on their projection in the PCA space. Two clustering methods were employed: graph-based and k-means. Cell Ranger also generated a table indicating the genes that exhibited differential expression in each cluster compared to all other clusters. The classification of PBMCs was inferred from the annotation of cluster-specific genes, relying on the expression of well-known markers of immune cell types (marker-based classification). For comprehensive visualization of the entire dataset and interactive exploration of significant genes, cell types, and substructure within cell clusters, Loupe Cell Browser (v5.0) was employed [31].

To process multiplexed samples, the HTO data are processed using the “cite-seq count, v1.4.3” tool available at https://hoohm.github.io/CITE-seq-Count/ (accessed on 5 January 2022) and https://zenodo.org/record/2590196 (accessed on 5 January 2022). The resulting data are then read using the Seurat “Read10X” tool, accessible at https://satijalab.org/seurat/articles/hashing_vignette.html (accessed on 5 January 2022). Only the cell barcodes that are common between the HTO data obtained from cite-seq count and the mRNA data obtained from cellranger are retained. These barcodes correspond to the cells detected by cellranger that also have an assigned HTO. Finally, the HTO arrays undergo normalization using the Log-Ratio Centered (CLR) method, which is recommended by the developers of Seurat. Subsequently, the “MULTIseqDemux” function is utilized to assign an HTO name to each cell.

To perform filtering on the multiplexed samples, the mRNA arrays obtained from cellranger count were loaded, and the cell barcodes were filtered based on the availability of an assigned HTO using MULTIseqDemux.

### 2.4. Gene Ontology Analysis for Molecular Function

Gene Ontology analysis for molecular function was performed using Panther version 16.0 database http://pantherdb.org (accessed on 6 June 2022)/.

### 2.5. Code Availability

All codes used in this study are available online at GitHub—SebastienNin/mw-Madaci2021.

## 3. Results and Discussion

### 3.1. Transcript and Gene Numbers from Single Cell RNA-seq and Cell Hashing Are Similar

In order to investigate the impact of multiplexing samples on the transcriptome through the utilization of a cell hashing technique, we employed peripheral blood mononuclear cells (PBMCs) obtained from two patients diagnosed with acute myeloid leukemia (AML). These cells were labeled with antibodies that targeted a protein expressed on the surface of all cells, coupled with a hashtag oligonucleotide (HTO). Subsequently, the labeled cells were mixed together prior to sequencing.

The multiplexing single cell RNA-seq technique, also known as cell hashing, relies on labeling ubiquitous membrane proteins with HTOs. In the context of human cells, the B2M and ATP1B3 proteins, which are widely expressed in normal cells, have been chosen as targets for cell labeling. However, interactions between extracellular molecules and membrane proteins can potentially impact various metabolic pathways. It raises concerns about whether such interactions could alter the transcriptome of leukemia cells. To address this, we examined the expression of the B2M protein in the two acute myeloid leukemia (AML) samples sequenced using single cell RNA-seq. The results of this analysis revealed high expression of the B2M protein in both AML samples, indicating that the process of leukemogenesis did not affect the expression of this cellular target (data not shown). With the confirmation of our HTO target, we proceeded to sequence the two AML samples using both techniques.

We utilized 10X Genomics scRNA-seq technology to sequence a total of 4390 cells from the UPN23 Single Cell RNA-sequencing (UPN23 SC) sample and 16,645 cells from the UPN29 SC sample, each sequenced individually. Additionally, we employed the cell hashing technique to sequence 5601 cells and 6480 cells in the UPN23 SC-m (single cell multiplexed) and UPN29 SC-m samples, respectively. The UPN23 SC sample exhibited a median of approximately 1660 genes per cell (median 1603; 1st–3rd quartile, 2248–2972), whereas the UPN23 SC-m sample showed a median of approximately 978 genes per cell (median 949; 1st–3rd quartile, 1510–2007). In the case of the UPN29 sample, we detected approximately 2191 genes per cell in UPN29 SC (median 1373; 1st–3rd quartile, 2209–2759) compared to 1763 genes per cell in UPN29 SC-m (median 1356; 1st–3rd quartile, 1772–2153) (Figure 1a, Table 2). The number of unique molecular identifiers (UMIs) captured, which represents the transcripts identified in each sequence read to prevent inflation due to PCR amplification during library preparation, was relatively higher in the individually sequenced samples (median of 4015 for UPN23 SC and 7765 for UPN29 SC) compared to the multiplexed samples (median of 3978 in UPN23 SC-m and 5554 in UPN29 SC-m) (Figure 1b). Furthermore, we observed consistency between the two approaches in terms of the percentages of mitochondrial genes in each patient (Figure 1c). Based on this quality control analysis, we found consistency in the sequencing results obtained from both approaches.

After applying cell filtering, we identified approximately 2579 cells in UPN23 SC compared to 3176 cells in UPN23 SC-m. Similarly, we detected approximately 14,722 cells in UPN29 SC and 6347 cells in UPN29 SC-m (Figure 1d,e). Subsequently, we performed clustering analysis on the cells from each sample using Seurat software. In UPN23 SC, we identified 12 distinct clusters labeled as clusters 0–11, whereas in UPN23 SC-m, we observed 11 clusters labeled as clusters 0–10 (Appendix A). As for the UPN29 sample, we discovered 16 clusters labeled as clusters 0–15 in UPN29 SC, and 13 clusters labeled as clusters 0–12 in UPN29 SC-m (Appendix A).

Next, we proceeded to assign cell subtypes to the cells using the bioinformatic program “celldex”. The results of this analysis enabled us to classify the cells from each sample into different subtypes, including leukemic cells, B cells, CD4+ T cells, CD8+ T cells, and NK lymphocytes. In UPN23SC, we identified approximately 95% leukemic cells, while in UPN23SC-m, we observed approximately 93% leukemic cells (Figure 1a). Similarly, in UPN29SC, we found approximately 90% leukemic cells, and in UPN29SC-m, we detected around 92% leukemic cells (Figure 1b). These findings indicate a strong correlation between the two approaches.

### 3.2. Gene Expression Levels Correlate between scRNA-seq and Cell Hashing

To complement our initial analysis, we investigated the correlation between the average gene expression in samples sequenced individually and those sequenced using the cell hashing technique. As depicted in Figure 1f–g, the expression levels of genes in both the UPN23 and UPN29 samples exhibited a strong correlation when comparing the two approaches (R = 0.996, *p* < 2.2 × 10^−16^ for UPN23 and R = 0.998, *p* < 2.2 × 10^−16^ for UPN29). 

In summary, our data strongly indicate that the cell hashing approach effectively maintains the integrity of individual cell transcriptomes, making it a reliable method for investigating gene expression.

Following the quality control of the two samples sequenced using different approaches, our focus shifted to the analysis of genes utilized as markers for diagnosing leukemic blasts (Table 1). Specifically, we examined the expression of KIT (CD117), CD34, CD33, CD13, CD36, and CD64 in UPN23 patients sequenced individually and with the cell hashing technique (Figure 2a). The results demonstrated a positive correlation in the expression of these genes between UPN23 SC and UPN23 SC-m (Figure 2b). Similarly, we conducted the same analysis for UPN29, investigating the expression of KIT (CD117), CD34, CD33, and CD11 as leukemic blast markers (Figure 2c). Once again, a correlation was observed in the expression of these genes between UPN29 SC and UPN29 SC-m (Figure 2d). These findings highlight the concordance in the expression of leukemic blast markers across both approaches, despite a reduction in the number of genes and gene expression variability in the cell hashing approach compared to the standard single cell approach.

Following the examination of leukemic markers, we proceeded to investigate the cluster-specific genes in both samples. To achieve this, we performed an analysis of differentially expressed genes for each sample and approach. The outcomes of this analysis are depicted in a heatmap (Appendix A), where each cluster is represented, along with the top overexpressed genes for each condition.

The analysis of these genes revealed no differences in gene expression between UPN23 SC (Appendix A) and UPN23 SC-m (Appendix A). Specifically, we observed consistent expression levels of cell markers such as MS4A1, CD79A and B, CD3D, NKG7, GZMA, and GNLY, which are indicative of normal hematopoietic cells including B lymphocytes, T lymphocytes, and natural killer (NK) cells. Additionally, we observed similar expression patterns of genes involved in the leukemic process, such as MPO and SOX4. Similarly, in UPN29 SC and UPN29 SC-m (Appendix A), we observed consistent expression of markers for leukemic blasts (LB), lymphoid cells (LT), and NK cells (CD79A, NKG7, and GNLY), as well as genes implicated in leukemogenesis (MPO and SOX4).

The analysis of the top 10 marker genes expressed in each cluster of UPN23 and UPN29 revealed a substantial overlap. Specifically, we found 93 common top 10 marker genes between UPN23 SC and UPN23 SC-m (Figure 3a), and approximately 104 common top 10 marker genes between UPN29 SC and UPN29 SC-m (Figure 3b). This analysis demonstrates the consistent recovery of marker genes with both approaches. This consistency was further supported by Gene Ontology analysis of molecular function performed on the top 10 most expressed genes in each cluster of UPN23 (Figure 3c) and UPN29 (Figure 3d). The results revealed that these genes could be categorized into six different functional categories, including transporter activity, structural molecular activity, molecular transducer activity, molecular function, catalytic activity, and binding. These findings highlight the agreement between the two approaches and provide evidence for the reliability of the cell hashing technique, which produces results comparable to those obtained through single cell RNA-seq.

## 4. Conclusions

In summary, our comparative study of single cell RNA sequencing methods reveals that the cell-hopping technique yields favorable outcomes while preserving transcriptomic information. We observed a strong correlation between gene expression and cellular markers, surpassing the traditional single cell approach. By employing cell labeling with HTOs, the cell-hopping approach allows for the sequencing of multiple samples in a single run, enabling a straightforward comparison between different samples and conditions without concerns about batch effects.

However, caution is advised when applying this approach, especially when identifying rare cell populations. As the number of samples increases, the number of sequenced cells per sample decreases, limiting the detection of infrequent cell populations. Based on our analysis capacity of approximately 30,000 cells per experiment, we find that using eight simultaneous samples is optimal in most cases. This allows for the analysis of relatively rare populations within each sample, even at low frequencies (e.g., 1%), which would still comprise around 40 cells using this technique. For smaller populations, the traditional single cell analysis would be more suitable.

Furthermore, a multiplexed single cell transcriptomic study has demonstrated its potential in assessing the sensitivity of leukemic cells to chemotherapy while elucidating underlying mechanisms [40]. Integrating the cell hashing technique with genetic and phenotypic approaches will facilitate the identification of distinct leukemic cell sub-populations, the discovery of new diagnostic and prognostic markers, and the establishment of targeted therapies for personalized clinical management of acute myeloid leukemia patients [41].

## Figures and Tables

**Figure 1 diseases-11-00096-f001:**
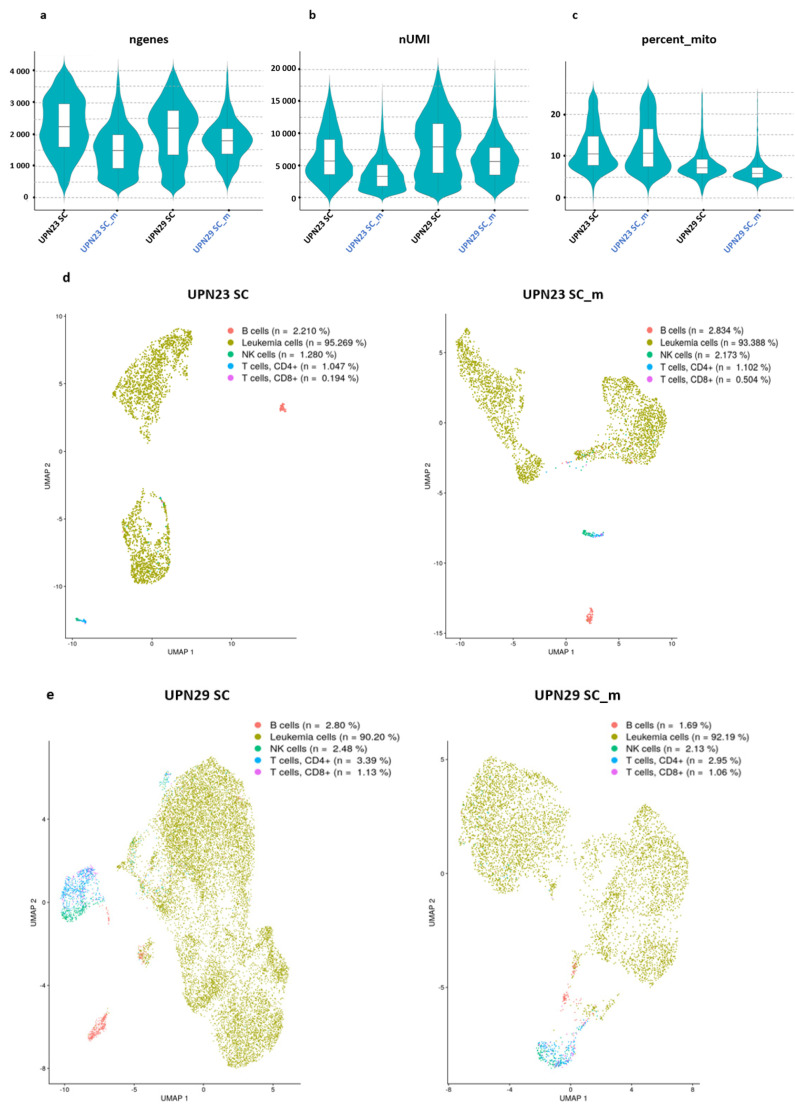
Single Cell transcriptome analysis of peripheral blood mononuclear cells (PBMCs) from acute myeloid leukemia (AML) patients. The figures present various quality control (QC) metrics for AML cells sequenced using both the single cell and cell hashing approaches. (**a**) A violin plot illustrates the number of genes detected, (**b**) the count of unique molecular identifiers (UMIs), and the percentage of mitochondrial genes (**c**) in each cell for the UPN23 SC (Single Cell RNA-sequencing) versus UPN23 SC-m (Single Cell Multiplexed) samples and the UPN29 SC versus UPN29 SC-m samples. (**d**) UMAPs (Uniform Manifold Approximation and Projection) display the cell clusters for 4390 cells from UPN23 SC (left) and 5601 cells from UPN23 SC-m (right), with distinct colors indicating different clusters. (**e**) UMAPs present the cell clusters for 16,645 cells from UPN29 SC (left) and 6480 cells from UPN29 SC-m (right), with colors representing different clusters. (**f**,**g**) Pairwise Pearson linear correlation analysis demonstrates the mean gene expression correlation between UPN23 SC and UPN23 SC-m (**f**), and UPN29 SC versus UPN29 SC-m (**g**).

**Figure 2 diseases-11-00096-f002:**
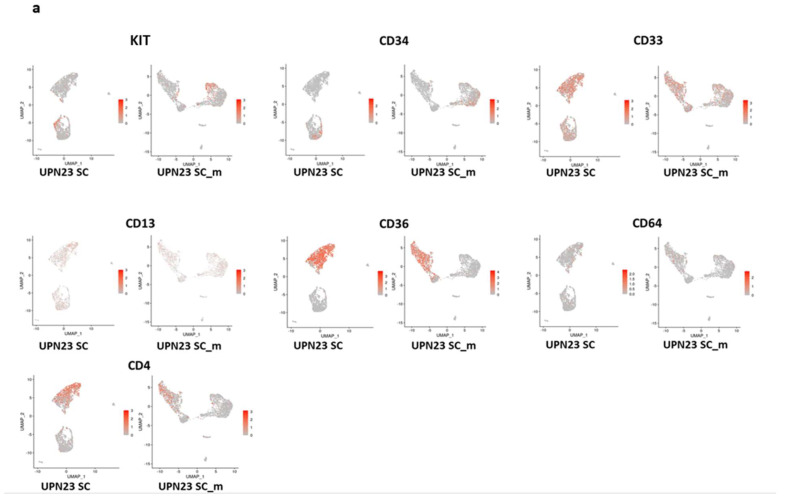
Identification of distinct leukemic cell markers specific to each sample. (**a**) UMAP visualization showing comparison of leukemia cell marker expression (KIT, CD34, CD33, CD13, CD36, CD64 and CD4) between UPN23 SC (Single Cell RNA-sequencing) and UPN23 SC_mSC-m (Single Cell-multiplexed). (**b**) Histogram illustrating the percentage of expression for these markers in UPN23 SC and UPN23 SC-m. (**c**) UMAP representation showing comparison of leukemia cell marker expression (KIT, CD34, CD33 and CD11) between UPN29 SC and UPN29 SC-m. (**d**) Histogram displaying the percentage of expression for these markers in UPN29 SC and UPN29 SC-m.

**Figure 3 diseases-11-00096-f003:**
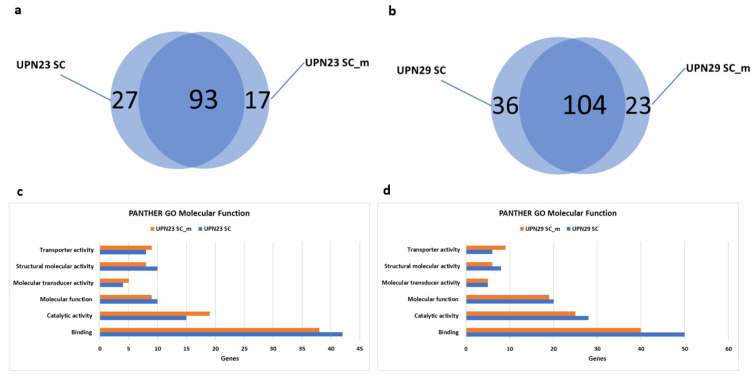
Comparison of top 10 expressed genes in each cluster of each sample. (**a**,**b**) Venn diagram illustrating the overlap of the 10 most expressed marker genes in clusters of UPN23 (**a**) and UPN29 (**b**). The diagram reveals 93 shared genes between UPN23 SC (Single Cell RNA-sequencing) and UPN23 SC-m (Single Cell_multiplexed) (left), and 104 shared genes between UPN29 SC and UPN29 SC-m. (**c**,**d**) Bar chart presenting the Panther Gene Ontology Molecular Function analysis of the top 10 up-expressed genes in each cluster of UPN23 (**c**) and UPN29 (**d**). The genes were classified into six molecular functions, including transporter activity, structural molecular activity, molecular transducer activity, molecular function, catalytic activity, and binding.

**Table 1 diseases-11-00096-t001:** Clinical characteristic of the samples.

Sample	UPN23	UPN29
Sex	F	M
Age	62	47
Nature of samples	Blood	Blood
% Blasts in blood	92%	83%
Cytology	AML4	AML4
Prognostic group	Adverse	Favorable
Phenotype	CD117, CD13, CD33, CD34 (myeloid blasts) CD64, CD4low, CD33 fort, CD36 (monocytic blasts)	CD117, CD34, CD33, CD11.
Cytogenetic	46XX, t(9;22)(p22;q23)	46XY, inv(16)
Genetic abnormalities	FLT 3—ITD, EVI1 et WT1	CBFβ

UPN: Unique Patient Number; F: Female; M: Male; AML: Acute Myeloid Leukemia; CD: Cluster of Differentiation; t: translocation; inv: inversion; FLT3: Fms-Like receptor Tyrosine kinase class III; ITD: Internal Tandem Duplication; EVI 1: Ecotropic Viral Integration site 1; WT 1: Wilms Tumor 1; CBFβ: Core-Binding Factor Subunit beta.

**Table 2 diseases-11-00096-t002:** Sequencing statistics.

Sample	Estimated Number of Cells	Total Read Number	Mean Reads per Cell	Median Genes per Cell
UPN23 SC	4390	171,943,491	39,167	1660
UPN23 SC-m	5601	207,650,556	37,100	978
UPN29 SC	16,645	711,710,869	42,758	2191
UPN29 SC-m	6480	341,065,053	52,633	1763

UPN: Unique Patient Number; SC: Single Cell RNA-seq; SC-m: single cell_multiplexed.

## Data Availability

The snRNA-seq data was downloaded from NCBI Gene Expression Omnibus (GEO) public database (GSE212038).

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
