# Peer review of "The Contribution of Multiplexing Single Cell RNA Sequencing in Acute Myeloid Leukemia"

_diseases, 2023, doi:10.3390/diseases11030096_

Round 1
Reviewer 1 Report
There is not much novelty in this manuscript. The multiplexed technique using Cell Hashing has already been well established, and thus the results are anticipated. The manuscript does not provide novel insight for the reader of AML single-cell analysis.
Minor point: In Table 1, the description of the Molecular Biology section is unclear. Is it referring to the mutation of the mentioned genes? Additionally, it is a bit surprising that the blast percentage in UPN29 is only about 83% while the single-cell sequencing result shows the percentage of leukemia cells to be over 90% in this sample.
na
Author Response
Minor point: In Table 1, the description of the Molecular Biology section is unclear. Is it referring to the mutation of the mentioned genes? è Molecular Biology means gene mutations in patients, as it's not clear, I've changed it to Genetic abnormalities.
Additionally, it is a bit surprising that the blast percentage in UPN29 is only about 83% while the single-cell sequencing result shows the percentage of leukemia cells to be over 90% in this sample.
At UPN29, the percentage of blasts was 83% in whole blood, while 90% of blasts corresponded to the percentage of blasts in PBMCs after separation with Ficoll.
Reviewer 2 Report
Comments for Madaci et al.
Major points:
1. Limitation of this study is the use of only 2 AML patient samples. Please explain if there is a value to including at least 1 healthy donor in this study.
2. The false negative rates for Cell Hashing, if I understand correctly, would be the true single cells that do not receive sufficient Cell Hashing signal. Please indicate would there be any false-negative rate associated with the Cell Hashing method for this study.
3. The Cell Hashing technique is to detect makers that are expressed in the patient samples. Should we caution that there can be in stances when the sample of interest does not express a particular diagnostic or prognostic marker?
4. Molecular markers that predict higher rates of relapse include FLT3 internal tandem duplications, mutations of RUNX1, ASXL1, and TP53 genes, or deregulated expression of the EVI1 gene. Please explain how the Cell Hashing technique would be beneficial to address this issue.
Minor points:
1. In addition to denoting via notations such as SC and SC_m, Ppease label in all the figures clearly which samples have undergone Cell Hashing and single cell sequencing.
2. Please indicate the interquartile range, and median for the violin plots in figure 1 (a to c).
3. Please indicate the P value for the correlation analysis for figure 1(f and g).
Author Response
Major points:
- Limitation of this study is the use of only 2 AML patient samples. Please explain if there is a value to including at least 1 healthy donor in this study.
è Although the sample size for this study was limited, we sought to assess the feasibility and reliability of the cell hashing technique on PBMCs from patients with acute myeloid leukemia, referring to the 2018 study by Stoeckius et al ( https://doi.org/10.1186/s13059-018-1603-1). Given that one study has already been carried out on healthy donor samples, we decided to use available funds to set up further studies, given the high cost of single-cell scale manipulations.
- The false negative rates for Cell Hashing, if I understand correctly, would be the true single cells that do not receive sufficient Cell Hashing signal. Please indicate would there be any false-negative rate associated with the Cell Hashing method for this study.
è In the cell hashing technique, we don't talk about false negatives, but rather about the rate of unlabeled cells. Even with the use of labeling oligonucleotides (HTO), there is always a percentage of unlabeled cells. In our study, of all the cells detected by Cell Ranger after sequencing, around 20% were not assigned to an HTO. This 20% includes both unlabeled cells and cells doubly labeled with both HTOs.
- The Cell Hashing technique is to detect makers that are expressed in the patient samples. Should we caution that there can be in stances when the sample of interest does not express a particular diagnostic or prognostic marker?
è The Cell Hashing technique can be used to detect specific diagnostic or prognostic markers in patient samples. However, it is important to note that there are situations where the sample examined does not express a particular marker. This high-throughput sequencing approach uses distinct markers to identify and characterize different cell populations. The presence or expression of diagnostic or prognostic markers may vary from one sample to another due to factors such as disease stage, cell composition and biological variations. Consequently, some markers may not be universally present, and some samples may not display specific markers relevant to the diagnosis or prognosis of the disease under study.
4. Molecular markers that predict higher rates of relapse include FLT3 internal tandem duplications, mutations of RUNX1, ASXL1, and TP53 genes, or deregulated expression of the EVI1 gene. Please explain how the Cell Hashing technique would be beneficial to address this issue.
è The Cell Hashing technique can be a useful method for predicting relapse rates by identifying markers or gene signatures associated with this probability. However, a multidimensional approach and further validation are generally required to obtain more accurate and reliable predictions in the clinical context.
Minor points:
- In addition to denoting via notations such as SC and SC_m, Ppease label in all the figures clearly which samples have undergone Cell Hashing and single cell sequencing.
- Please indicate the interquartile range, and median for the violin plots in figure 1 (a to c).
- Please indicate the P value for the correlation analysis for figure 1(f and g).
I have incorporated the recommendations provided in points 1 and 3 into the manuscript. In addition, I have included medians and interquartile range for each sample and parameter in the second paragraph of the results section. In view of the potential information overload, I have decided not to include these values in Figures 1a-c.
Reviewer 3 Report
The authors conducted a thorough investigation into the impact of cell hashing by HTO on primary AML samples and successfully validated its performance. The efforts and data resource provided in this study are highly valuable. However, there are some areas that need improvement. 1) The writing and wording throughout the paper should be modified to improve clarity and flow. Consider using ChatGPT. 2) Certain details mentioned in the paper need to be filled in or corrected as stated below. 3) proper citations to literature, specifically studies utilizing scRNA-seq in AML, should be included. This will ensure that the authors acknowledge the existing body of research in the field. Specifically, the authors should cite relevant studies conducted by Ianis Aifantis lab, Vijay G Sankaran, Peter Van Galen, and others that are pertinent to AML scRNA-seq. 4) It is important to avoid making overstated claims about the immediate clinical use or the extension of target therapy based on the use of scRNA-seq with multiplexing. This paper should remain focused on the technical aspects of the study.
Line 26. “classic” single-cell analysis is not appropriate. Use “standard”, as opposed to “multiplexed”
Line 63, cite “CITE-seq” and other antibody-based scRNA-seq that can profile hundreds of antibodies at single-cell resolution.
Line 68-72, there are tons of AML studies and papers with scRNA-seq including these groups, Ianis Aifantis, Vijay G Sankaran, Peter Van Galen etc. Please examine the literature carefully and cite them properly.
Line 83, extra space
*Line 150-159, recommended sequencer setup for 3’ V3 GEX is 28+8+0+91. The authors said they used 55bp for R2 which is incorrect. Please check with your people and re-write this method section. Also, include how you sequenced the HTO libraries and sequencing depth.
Line 309, “genetic information or transcriptomic modification” change to “transcriptomic information”
Line 319, the authors need to provide a sampling power calculation. There are many tools available for example: https://www.ncbi.nlm.nih.gov/pmc/articles/PMC6852764/
*Line 326-332, there are many unsolved questions and missing links between the method (scRNA-seq with hashing) and potential clinical use. Please examine the literature and address existing papers.
Figure 2a, do not use protein names eg.CD117 when you are displaying RNA. This is misleading since CITE-seq was not performed. Just use the gene symbol. Also, there is a known disconnect between mRNA expression and protein, so careful with the claims. It would be helpful to compare the frequency of cells that are positive for each/combination of these markers by flow vs those that have RNA expression.
Please use ChatGPT/ a scientific writer to rewrite this manuscript.
